# Investigation of Perovskite Solar Cells Using Guanidinium Doped MAPbI_3_ Active Layer

**DOI:** 10.3390/nano14080657

**Published:** 2024-04-10

**Authors:** Ting-Chun Chang, Ching-Ting Lee, Hsin-Ying Lee

**Affiliations:** 1Department of Photonics, National Cheng Kung University, Tainan 701, Taiwan; l78111508@gs.ncku.edu.tw (T.-C.C.); ctlee@ee.ncku.edu.tw (C.-T.L.); 2Institute of Microelectronics, Department of Electrical Engineering, National Cheng Kung University, Tainan 701, Taiwan; 3Department of Electrical Engineering, Yuan Ze University, Taoyuan 320, Taiwan; 4Meta-nanoPhotonics Center, National Cheng Kung University, Tainan 701, Taiwan

**Keywords:** crystallinity and crystal grain size, guanidinium-doped methylammonium lead triiodide, perovskite solar cells, surface morphology, X-ray diffraction

## Abstract

In this work, guanidinium (GA^+^) was doped into methylammonium lead triiodide (MAPbI_3_) perovskite film to fabricate perovskite solar cells (PSCs). To determine the optimal formulation of the resulting guanidinium-doped MAPbI_3_ ((GA)_x_(MA)_1−x_PbI_3_) for the perovskite active layer in PSCs, the perovskite films with various GA^+^ doping concentrations, annealing temperatures, and thicknesses were systematically modulated and studied. The experimental results demonstrated a 400-nm-thick (GA)_x_(MA)_1−x_PbI_3_ film, with 5% GA^+^ doping and annealed at 90 °C for 20 min, provided optimal surface morphology and crystallinity. The PSCs configured with the optimal (GA)_x_(MA)_1−x_PbI_3_ perovskite active layer exhibited an open-circuit voltage of 0.891 V, a short-circuit current density of 24.21 mA/cm^2^, a fill factor of 73.1%, and a power conversion efficiency of 15.78%, respectively. Furthermore, the stability of PSCs featuring this optimized (GA)_x_(MA)_1−x_PbI_3_ perovskite active layer was significantly enhanced.

## 1. Introduction

Global warming has led to numerous fatalities and severe global issues. The culprit of the tragedy is the excessive use of fossil fuel [1]. Therefore, the demand for clean, pollution-free, and sustainable energy sources is imminent. Among various green energy options, solar cells, particularly favored due to advancements in related technologies, emerge as a prime alternative to fossil fuels [2,3,4,5]. Nowadays, the research on organic-inorganic metal halide perovskite solar cells (PSCs) has shown excellent progress around the world [6,7,8,9,10]. Organic perovskite, a type of organic semiconductors with an ABX_3_ lattice structure, where A represents the organic cation, B represents the metal cation, and X represents the halogen anion, are notable in photovoltaics for their versatility, simple process, tunable bandgap, high carrier mobility, and so on [11,12,13]. Remarkably, the power conversion efficiency (PCE) of PSCs has increased from 3.8% to 26% in a short period [14]. This rapid development has attracted wide attention and involvement in the field [15,16]. However, because the performance and stability of the perovskite are easily deteriorated due to the penetration of water and oxygen molecules, it poses challenges for commercialization [17]. Hence, achieving high PCE and stability in PSCs remains a critical global research topic [18,19]. Since doping technology in perovskite has been discovered, it represents a wonderful promising solution to enhance the performance and stability of PSCs [20,21].

Because guanidinium (GA^+^) has a slightly larger organic cation volume than that of methylammonium (MA^+^), it is regarded as an important material to form the inorganic-organic low-dimensional perovskite crystal. The resulting perovskite exhibits better stability than the common three-dimensional perovskite such as methylammonium lead triiodide (MAPbI_3_) [22,23]. Additionally, GA^+^ has been extensively investigated as an additive agent to improve the performance and stability of dye-sensitized solar cells (DSSCs) [24]. Jeanbourquin et al. successfully applied GA^+^ into the electrolyte of DSSCs to reduce the recombination and lower the conduction band energy, which enhanced the short-circuit current density of the DSSCs by 35% [25]. However, the excessive low-dimensional perovskite compounds would seriously impair the carrier’s vertical transmission ability, thus reducing PSC performance [26]. Based on these unique perovskite properties, doping technology emerges as a promising method to improve performance and stability of PSCs [27,28]. For example, Liu et al. successfully doped N,1-fluoroformamidine (F-FA^+^) into MAPbI_3_ perovskite to form the F-FA/MAPbI_3_ perovskite active layer and achieve a great PCE of 17.01% [29]. Consequently, in this work, to obtain the suitable GA^+^ doping concentration for the perovskite active layer, guanidinium iodide (C(NH_2_)_3_I, GAI) and methylammonium iodide (CH_3_NH_3_I, MAI) with varying molar ratios were mixed with lead iodide (PbI_2_), forming guanidinium-doped MAPbI_3_ ((GA)_x_(MA)_1−x_PbI_3_) perovskite crystal, where x represents the GA^+^ doping concentration. Subsequently, different annealing temperatures and thicknesses of (GA)_x_(MA)_1−x_PbI_3_ active layer were also investigated to achieve the optimal (GA)_x_(MA)_1−x_PbI_3_ PSCs.

## 2. Materials and Methods

### 2.1. Materials

In this work, indium tin oxide (ITO)-coated glass substrates, poly(3,4-ethylenedioxythiophene):poly(styrenesulfonate) (PEDOT:PSS) conductive solution (1.3–1.7 wt%), GAI powder (>99%), MAI powder (98%), fullerene C_70_ powder (98%), and bathocuproine (BCP) powder (>99.5%) were purchased from Uni-onward Corp., New Taipei City, Taiwan. PbI_2_ powder (99%) was purchased from Alfa Aesar, Haverhill, MA, USA. Dimethyl sulfoxide (DMSO) solvent (99.9%), γ- butyrolactone (GBL) solvent (99%), and chlorobenzene (CB) solvent (99.8%) were purchased from Sigma-Aldrich, St. Louis, MI, USA.

### 2.2. Manufacturing Process

The three-dimensional schematic configuration and the corresponding energy level diagram of the PSCs with (GA)_x_(MA)_1−x_PbI_3_ perovskite active layer are shown in Figure 1a,b, respectively. First, a patterned ITO-coated glass substrate was sequentially soaked in acetone, methanol, deionized water, and cleaned by an ultrasonic cleaner for 5 min. Next, a 50-nm-thick PEDOT:PSS hole transport layer (HTL) was spin-coated on the patterned ITO anode and annealed in a N_2_ ambient at 120 °C for 15 min. To form the (GA)_x_(MA)_1−x_PbI_3_ perovskite solution with different GA^+^ doping concentrations of 0, 5, 10, and 15%, various molar ratios of GAI powder and MAI powder were mixed with PbI_2_ (1.157 g) powder, DMSO solvent (1 mL), and GBL solvent (1 mL). Subsequently, the (GA)_x_(MA)_1−x_PbI_3_ perovskite solution was spun on the PEDOT:PSS HTL and annealed in a N_2_ ambient at different temperatures of 70, 90, 110, and 130 °C for 20 min to form (GA)_x_(MA)_1−x_PbI_3_ perovskite active layer. Besides, (GA)_x_(MA)_1−x_PbI_3_ film with the thicknesses of 300, 400, and 500 nm could be obtained by controlling the rotational speed. Finally, a 30-nm-thick fullerene C_70_ electron transport layer (ETL), a 10-nm-thick BCP hole-blocking layer, and a 100-nm-thick Ag cathode electrode were sequentially evaporated on the (GA)_x_(MA)_1−x_PbI_3_ active layer by a thermal evaporator.

The surface morphologies of the various (GA)_x_(MA)_1−x_PbI_3_ films with different GA^+^ doping concentrations and annealing temperatures were observed by a field emission scanning electron microscope (FE-SEM, AURIGA, ZEISS, Oberkochen, Germany). The crystallinity of the various (GA)_x_(MA)_1−x_PbI_3_ films were characterized by a grazing incidence X-ray diffraction system (GIXRD, AXS Gmbh, Bruker, Billerica, MA, USA). The roughness of the various-thickness-formed (GA)_x_(MA)_1−x_PbI_3_ films was measured using an atomic force microscope (AFM, Dimension ICON, Bruker, Billerica, MA, USA). The optical transmission of the various (GA)_x_(MA)_1−x_PbI_3_ films was measured using a UV–Visible–NIR spectrophotometer (U-4100, HITACHI, Tokyo, Japan). The current density-voltage (J-V) characteristics of the various perovskite solar cells were measured using a Keithley 2400 (Keithley Instruments, Solon, OH, USA) under an AM1.5G solar simulator (100 mW/cm^2^) (Forter Technology Corp., Taichung, Taiwan). The spectral external quantum efficiency (EQE) of the various perovskite solar cells were measured using an Xe lamp source with a power of 150 W and a monochromator (QE-3000, Zolix, Beijing, China).

## 3. Experimental Results and Discussions

To achieve high-quality films, various GA^+^ doping concentrations of 0, 5, 10, and 15% were doped to obtain (GA)_x_(MA)_1−x_PbI_3_ films. The various 300-nm-thick (GA)_x_(MA)_1−x_PbI_3_ films were then annealed in a N_2_ ambient at 90 °C for 20 min. The XRD patterns, as shown in Figure 2, revealed the predominant (110) lattice plane of (GA)_x_(MA)_1−x_PbI_3_ films at 14.06°. As the GA^+^ doping concentration increased from 0% to 5%, a notable increase in the (110) peak intensity was observed at 5% GA^+^ doping concentration. This phenomenon was attributed to the fact that the GA^+^ had a better hydrogen bonding capability to construct a stronger perovskite crystal, which could enhance the crystallinity of the (GA)_x_(MA)_1−x_PbI_3_ films and increased the intensity of the (110) diffraction peak [30]. However, the intensity of the (110) diffraction peak attenuated as the doping concentration further increased to 10%. This attenuation was attributed to the fact that excess low-dimensional (GA)_x_(MA)_1−x_PbI_3_ perovskite crystals, generated by doping excess GA^+^, deteriorated the crystallinity of the original three-dimensional (GA)_x_(MA)_1−x_PbI_3_ film and reduced the carrier vertical transmission ability [31,32,33]. As the GA^+^ doping concentration further increased to 15%, it led to the generation of a new diffraction peak at 11.3°, indicating low-dimensional crystal growth during excess GA^+^-doped. Consequently, when the GA^+^ doping concentration was 5%, the (GA)_x_(MA)_1−x_PbI_3_ film exhibited the best crystallinity.

The SEM images shown in Figure 3 were used to investigate the surface morphology of the various (GA)_x_(MA)_1−x_PbI_3_ films. According to the SEM images, compared with the one without GA^+^ doping, the (GA)_x_(MA)_1−x_PbI_3_ films with a GA^+^ doping concentration of 5% displayed a larger crystal grain size. The phenomenon was attributed to the superior hydrogen bonding of GA^+^, which was evident from Figure 3a,b. Conversely, as shown in Figure 3c, aligning with XRD results, because the excess GA^+^ doping concentration deteriorated the crystallinity of the original three-dimensional perovskite, the crystal grain size of the (GA)_x_(MA)_1−x_PbI_3_ films was significantly decreased as the GA^+^ doping concentration increased to 10%. Moreover, according to Figure 3d, when the GA^+^ doping concentration further increased to 15%, a significant low-dimensional (GA)_x_(MA)_1−x_PbI_3_ perovskite crystal appeared on the surface. It also verified the GA^+^ indeed generated the low-dimensional (GA)_x_(MA)_1−x_PbI_3_ perovskite crystal, which corresponded to the diffraction peak at 11.3° shown in Figure 2. Consequently, the (GA)_x_(MA)_1−x_PbI_3_ films with GA^+^ doping concentration of 5% had the best surface morphology and crystallinity, which promoted the carrier transportation ability and enhanced performances of the resulting PSCs devices. The results from SEM images corresponded to the trend of XRD analysis. As a result, the (GA)_x_(MA)_1−x_PbI_3_ films with an optimal GA^+^ doping concentration of 5% was defined as (GA)_0.05_(MA)_0.95_PbI_3_.

The influence of various annealing temperatures of 70, 90, 110, and 130 °C on (GA)_0.05_(MA)_0.95_PbI_3_ films was also explored in this work. Figure 4a shows the XRD results of the 300-nm-thick (GA)_0.05_(MA)_0.95_PbI_3_ films annealed at various temperatures. Using the full width at half maximum (FWHM) of the (110) diffraction peak shown in Figure 4a, the crystal grain size (D) of the various (GA)_0.05_(MA)_0.95_PbI_3_ films could be calculated and shown in Figure 4b by the following Equation (1) [34]: (1)$$D=\frac{K\lambda }{\beta \cos\theta }$$
where K represents the Scherrer constant, λ represents the wavelength of the X-ray, β represents the FWHM value of the main diffraction peak, and θ represents the diffraction angle. Each XRD and related crystal grain size analysis of the (GA)_0.05_(MA)_0.95_PbI_3_ films with various annealing temperatures had been measured over five times. The error bar presented in Figure 4b showed the statistical results of the XRD measurement. From the XRD results and the related crystal grain size presented in Figure 4a,b, when the annealing temperature increased from 70 °C to 130 °C, the crystallinity of the (GA)_0.05_(MA)_0.95_PbI_3_ films was enhanced and the associated crystal grain size was gently increased from 21.58 nm to 23.80 nm. This phenomenon was attributed to the fact that higher annealing temperature accelerated the formation kinetics of perovskite crystal, which could produce a perovskite film with better crystallinity and larger crystal grain size [35,36]. The larger crystal grain size was beneficial for the carrier transportation in the (GA)_0.05_(MA)_0.95_PbI_3_ films. However, along with the increasing annealing temperature, there was a diffraction peak gradually appearing at the angle of 12.56°, representing the (001) crystal plane of PbI_2_ molecules which originated from the degraded (GA)_0.05_(MA)_0.95_PbI_3_ perovskite crystal. The increasing annealing temperature also elevated the (001) diffraction peak intensity of PbI_2_ molecules. This phenomenon was attributed to the fact that the higher annealing temperature accelerated the degradation of (GA)_0.05_(MA)_0.95_PbI_3_, which tended to decompose into PbI_2_, MAI, and GAI molecules. It could easily generate pinholes and cracks on the perovskite surface. Consequently, the carrier transportation ability of the perovskite films deteriorated [37]. SEM surface analysis was used to further observe the surface morphology of the (GA)_0.05_(MA)_0.95_PbI_3_ films treated with various annealing temperatures. From the SEM images shown in Figure 5a–d, as the annealing temperature increased from 70 °C to 130 °C, the crystal grain size of the (GA)_0.05_(MA)_0.95_PbI_3_ films was indeed significantly increased due to the generation of higher formation kinetics, which could correspond to the results of XRD analysis in Figure 4. However, the higher annealing temperature also led to a higher decomposition rate. Consequently, pinholes and cracks were caused on the (GA)_0.05_(MA)_0.95_PbI_3_ films. As shown in Figure 5c, when the annealing temperature was 110 °C, pinholes and cracks appeared among the crystal boundaries of the annealed perovskite films. Compared to the (GA)_0.05_(MA)_0.95_PbI_3_ film annealed at a temperature of 110 °C, in addition to their larger crystal grain size, it also caused more pinholes and cracks residing on the perovskite films annealed at 130 °C, as shown in Figure 5d. Combining the results of XRD analysis and SEM images, when the annealing temperature of the (GA)_0.05_(MA)_0.95_PbI_3_ films was higher than 110 °C, many pinholes and cracks started to appear on the perovskite surface. It would seriously deteriorate the carrier transportation ability of the (GA)_0.05_(MA)_0.95_PbI_3_ perovskite films. To manufacture high-performance PSCs with an optimal (GA)_0.05_(MA)_0.95_PbI_3_ film, the best compromise among a larger perovskite crystal grain size, a lower decomposition rate, and less pinholes and cracks was needed to be investigated. Thus, to balance the crystal grain size, decomposition rate, and surface integrity, the temperature of 90 °C was identified as the optimal annealing temperature for fabricating high-performance PSCs in this work.

Figure 6a,b show the XRD patterns and the relating crystal grain size of the 300-nm-thick (GA)_0.05_(MA)_0.95_PbI_3_ films annealed for various times of 10, 20, and 30 min. The XRD patterns, as shown in Figure 6a, revealed the predominant (110) lattice plane of (GA)_0.05_(MA)_0.95_PbI_3_ films at 14.06°. According to the results, as the annealing time increased from 10 min to 20 min, the crystallinity of the (GA)_0.05_(MA)_0.95_PbI_3_ films was enhanced and the relating crystal grain size increased from 20.39 nm to 22.66 nm. This phenomenon illustrated under the annealing temperature of 90 °C, the annealing time of 10 min was not enough for obtaining the (GA)_0.05_(MA)_0.95_PbI_3_ films with high crystallinity. However, as the annealing time further increased from 20 min to 30 min, the diffraction peak of PbI_2_ molecules at 12.56° enhanced. This phenomenon was attributed to the fact that the excessive annealing time would also stimulate the degradation of (GA)_0.05_(MA)_0.95_PbI_3_ perovskite crystal and deteriorate the quality of (GA)_0.05_(MA)_0.95_PbI_3_ films. Besides, it did not present a significant enhancement of the relating crystal grain size of (GA)_0.05_(MA)_0.95_PbI_3_ films when the annealing time increased from 20 min to 30 min. The phenomenon showed if the annealing time exceeded 20 min, it would only stimulate the decomposition of (GA)_0.05_(MA)_0.95_PbI_3_ perovskite crystal but not generate the bigger (GA)_0.05_(MA)_0.95_PbI_3_ perovskite crystal grain size.

To study the impact of thickness variations in (GA)_0.05_(MA)_0.95_PbI_3_ films, ranging from 300 to 500 nm, the measured XRD results were shown in Figure 7. These various (GA)_0.05_(MA)_0.95_PbI_3_ films were annealed in a N_2_ ambient at 90 °C for 20 min. The crystallinity of various (GA)_0.05_(MA)_0.95_PbI_3_ films was found to be gradually elevated as the thickness increased from 300 nm to 500 nm. Based on the results in the stronger intensity of the (110) diffraction peak and the better crystallinity of the thicker (GA)_0.05_(MA)_0.95_PbI_3_ film, this above-mentioned phenomenon was attributed to the formation of denser perovskite molecules and multi-layered particle structures [38]. Because the better crystallinity and larger crystal grain size of the perovskite film led to increase carrier mobility within the polycrystalline structure, the performances of the resulting PSC devices were expected to be consequently enhanced. [39]. However, according to the results of AFM analysis presented in Figure 8a–c, the root-mean-square roughness (R_q_) of the (GA)_0.05_(MA)_0.95_PbI_3_ films increased from 8.50 nm to 10.50 nm and to 25.10 nm as the thickness of the films increased from 300 nm to 400 nm and to 500 nm. If the surface of the perovskite films was seriously roughened, more defects would be easily generated within the interface during the coverage of the following transport layer. The undesired defects could seriously deteriorate the carrier transportation ability of the resulting PSCs [40,41]. Consequently, the large roughness would bring some adverse influences on the performances of the (GA)_0.05_(MA)_0.95_PbI_3_ films. Figure 8d shows the related crystal grain size, calculated from Figure 7 and Equation (1), and R_q_ as a function of the thickness of the (GA)_0.05_(MA)_0.95_PbI_3_ films. Each XRD pattern and related crystal grain size analysis of the (GA)_0.05_(MA)_0.95_PbI_3_ films with various thicknesses had been measured over five times. The error bar presented in Figure 8d showed the statistical results of the XRD measurement. As above-mentioned crystal grain size and R_q_ of the (GA)_0.05_(MA)_0.95_PbI_3_ films, despite larger crystal grain sizes of the (GA)_0.05_(MA)_0.95_PbI_3_ film with a thickness of 500 nm being beneficial for carrier mobility, the associated excessive roughness would adversely affect the performance of the resulting PSCs. Accordingly, to determine the optimal thickness of the (GA)_0.05_(MA)_0.95_PbI_3_ film, an optimal balance between a larger perovskite crystal grain size and a smaller roughness was needed to be investigated. The performances of the resulting PSCs with various thicknesses of the (GA)_0.05_(MA)_0.95_PbI_3_ films were analyzed to ascertain the optimal thickness of the (GA)_0.05_(MA)_0.95_PbI_3_ perovskite active layer.

Figure 9 shows the absorption spectra of the various-thick (GA)_0.05_(MA)_0.95_PbI_3_ perovskite films. It was found the thicker perovskite films exhibited enhanced light absorption. The inserted figure in Figure 9 extended the absorption spectra from the wavelength of 700 nm to 800 nm. It demonstrated a significant change in light absorbance of the perovskite active layer with different thicknesses. When the thickness of the (GA)_0.05_(MA)_0.95_PbI_3_ perovskite film was 500 nm, it exhibited the highest light absorbance. The higher light absorbance of the perovskite active layer could absorb more photons to generate more carriers and enhance the short-circuit current density (J_sc_) of the resulting PSCs. Furthermore, because the better crystallinity caused in a thicker film could improve the carrier transport properties, the (GA)_0.05_(MA)_0.95_PbI_3_ perovskite active layer with increasing thickness could be expected to also enhance the J_sc_ of the resulting PSCs [42,43]. The current density-voltage, dark current density-voltage and external quantum efficiency (EQE) performances of the PSCs using the (GA)_0.05_(MA)_0.95_PbI_3_ perovskite active layer with a thickness of 300, 400, and 500 nm were illustrated in Figure 10a–c, respectively. The experimental results were also listed in Table 1. As shown in Figure 10a, in view of the higher light absorbance, better crystallinity, and larger crystal grain size, when the thickness of the (GA)_0.05_(MA)_0.95_PbI_3_ perovskite active layer increased from 300 nm to 400 nm, the J_sc_ of the resulting PSCs gradually increased from 23.88 mA/cm^2^ to 24.21 mA/cm^2^, which increased the PCE from 15.36% to 15.78%. However, although the J_sc_ of the PSCs could increase from 24.21 mA/cm^2^ to 24.75 mA/cm^2^ as the thickness of the (GA)_0.05_(MA)_0.95_PbI_3_ perovskite active layer further increased from 400 nm to 500 nm, the fill factor (FF) of the PSCs was relatively decreased from 73.1% to 66.6%. The deterioration in FF was attributed to the factor that the markedly increasing roughness of the 500-nm-thick (GA)_0.05_(MA)_0.95_PbI_3_ perovskite layer generated more defects at the interface of the (GA)_0.05_(MA)_0.95_PbI_3_ perovskite active layer and the subsequent fullerene C_70_ ETL. Consequently, the PCE of the resulting PSCs decreased from 15.78% to 14.73% as the thickness of the (GA)_0.05_(MA)_0.95_PbI_3_ perovskite active layer increased from 400 nm to 500 nm. Furthermore, the dark current density-voltage characteristics of PSCs using the various-thick (GA)_0.05_(MA)_0.95_PbI_3_ perovskite active layers were shown in Figure 10b. When the thickness of (GA)_0.05_(MA)_0.95_PbI_3_ perovskite active layer was 400 nm, it exhibited the lowest dark current. This was attributed to the phenomenon the better crystallinity could improve the carrier transport properties and the lower roughness could reduce the number of defects residing at the interface of the (GA)_0.05_(MA)_0.95_PbI_3_ perovskite active layer and the fullerene C_70_ ETL. On the basis of the illustration above, the optimal balance between the crystal grain size and the roughness occurred at the 400-nm-thick (GA)_0.05_(MA)_0.95_PbI_3_ perovskite active layer. The best performance of the resulting PSCs with an open-circuit voltage (V_oc_) of 0.891 V, J_sc_ of 24.21 mA/cm^2^, FF of 73.1%, and PCE of 15.78% were obtained. Since EQE is an important parameter for characterizing PSCs to assess their quality, Figure 10c shows the dependence of EQE on wavelength (300–800 nm) of the various PSCs. Besides, the integrated J_sc_ is also used to verify the value and trend of J_sc_ of the PSCs. Using the EQE spectra, integrated J_sc_ can be calculated by the following Equation (2) [44]: (2)$${\mathrm{Integrated}\;J}_{\mathrm{sc}}=\int \mathrm{EQE}\left(\lambda \right)\times \mathrm{qF}{\left(\lambda \right)}_{\mathrm{AM}1.5G}d\lambda$$
where F(λ) represents the wavelength (λ)-dependent incident photon flux density of AM 1.5G standard spectroscopy, EQE(λ) represents the wavelength-dependent external quantum efficiency, and q represents the electron charge. The integrated J_sc_ of the PSCs using the (GA)_0.05_(MA)_0.95_PbI_3_ perovskite active layer with the thicknesses of 300, 400, and 500 nm was 20.69, 21.55, and 22.21 mA/cm^2^, respectively. According to the results, although the PSCs using the 500-nm-thick (GA)_0.05_(MA)_0.95_PbI_3_ perovskite active layer revealed the highest EQE due to the higher light absorptance and the better crystallinity, the markedly increasing roughness would still deteriorate the FF and PCE performances of PSCs from 73.1% to 66.6% and from 15.78% to 14.73%, respectively. Synthesizing the aforementioned results, to achieve the best PCE performance, the optimal 400-nm-thick (GA)_0.05_(MA)_0.95_PbI_3_ perovskite active layer of PSCs was determined.

Figure 11a–c show the current density-voltage, dark current density-voltage and EQE performances of the PSCs with MAPbI_3_ perovskite active layer and with optimal (GA)_0.05_(MA)_0.95_PbI_3_ perovskite active layer, respectively. The relating experimental results were listed in Table 2. Compared with the one without GA^+^ doping, the PSCs with optimal (GA)_0.05_(MA)_0.95_PbI_3_ perovskite active layer showed a significant enhancement of device performance, which led to the V_oc_ being promoted from 0.783 V to 0.891 V, the J_sc_ increased from 19.71 mA/cm^2^ to 24.21 mA/cm^2^, the FF elevated from 65.0% to 73.1, and the PCE enhanced from 10.03% to 15.78%, respectively. The phenomenon was attributed to the fact that the better hydrogen bonding capability of GA^+^ could construct a stronger perovskite crystal and enhance the crystallinity of the perovskite active layer. Furthermore, after the optimization of doping concentration, annealing temperature and time, and thickness of (GA)_0.05_(MA)_0.95_PbI_3_ perovskite active layer, not only the device performance but also the stability of PSCs could be enhanced. Figure 11d illustrates the 200-h stability test of the PSCs with MAPbI_3_ perovskite active layer and with optimal (GA)_0.05_(MA)_0.95_PbI_3_ perovskite active layer, respectively. According to the results, after a 200-h stability test at the atmospheric environment with an ambient temperature of 25 °C and the relative humidity of 50%, the PSCs without GA^+^ only retained 42.6% of its original PCE, while 77.3% of the original PCE was retained by the PSCs with the optimal (GA)_0.05_(MA)_0.95_PbI_3_ perovskite active layer. This enhanced stability was attributed to GA^+^’s superior hydrogen bonding capability, which strengthens the crystal structure of the (GA)_0.05_(MA)_0.95_PbI_3_ perovskite active layer. Consequently, GA^+^ doping not only improved performance but also significantly boosted the long-term stability of the resulting PSCs.

## 4. Conclusions

In summary, based on the XRD and SEM analyses, owing to the GA^+^’s superior hydrogen bonding capability in constructing stronger perovskite crystals, the perovskite active film with a GA^+^ doping concentration of 5% was demonstrated as optimal film for superior crystallinity and surface morphology. The XRD and SEM results also demonstrated a higher annealing temperature was beneficial for better crystallinity and larger crystal grain size. However, this higher annealing temperature also increased the decomposition rate of the (GA)_0.05_(MA)_0.95_PbI_3_ films, simultaneously. By effectively balancing enhanced crystal growth against decomposition risks, the optimal annealing temperature and time was determined to be 90 °C and 20 min, respectively. Combining the results of XRD and AFM measurements, it was found the better crystallinity and the roughened texture of the (GA)_0.05_(MA)_0.95_PbI_3_ perovskite active layer were obtained as its thickness increased. When the thickness of the (GA)_0.05_(MA)_0.95_PbI_3_ films increased from 400 nm to 500 nm, the significantly induced defects caused by the serious roughness could deteriorate the performances of the resulting PSCs. Consequently, the 400-nm-thick ((GA)_0.05_(MA)_0.95_PbI_3_ perovskite active layer emerged as an optimal thickness. From the performance analysis of PSCs, when the thickness of the (GA)_0.05_(MA)_0.95_PbI_3_ perovskite active layer was 400 nm, it had optimal performance, with PCE, J_sc_, V_oc_, and FF of 15.78%, 24.21 mA/cm^2^, 0.891 V, and 73.1%, respectively. Notably, PSCs with the optimal (GA)_0.05_(MA)_0.95_PbI_3_ layer demonstrated significantly improved device performances, whose PCE increased from 10.03% to 15.78%. According to the experimental results, by doping GA^+^ into MAPbI_3_ perovskite films, not only performances improved but also long-term stability of the resulting (GA)_0.05_(MA)_0.95_PbI_3_ films, retaining 77.3% of their original PCE after stability test for 200 h compared with the 42.6% without GA^+^ additive doping. Consequently, using the optimal (GA)_0.05_(MA)_0.95_PbI_3_ perovskite active layers, the resulting PSCs with high performances and high stability have been successfully obtained.

## Figures and Tables

**Figure 1 nanomaterials-14-00657-f001:**
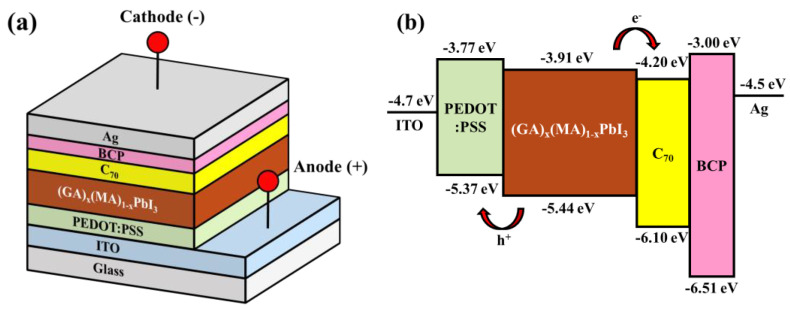
(**a**) Schematic configuration and (**b**) corresponding energy level diagram of perovskite solar cells with (GA)_x_(MA)_1−x_PbI_3_ perovskite active layer.

**Figure 2 nanomaterials-14-00657-f002:**
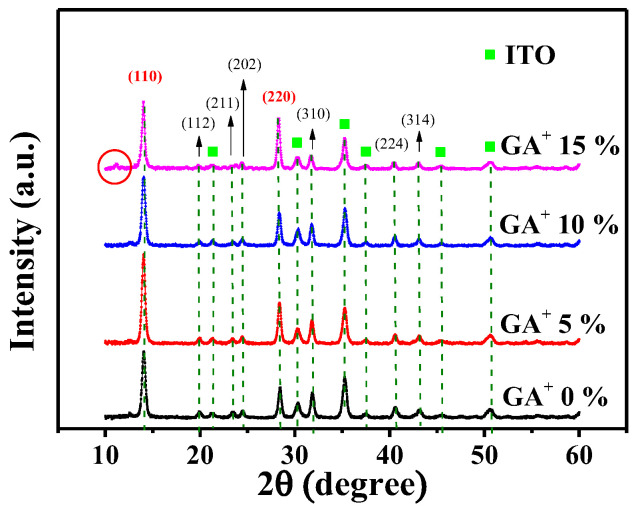
XRD patterns of (GA)_x_(MA)_1−x_PbI_3_ films with various GA^+^ doping concentrations.

**Figure 3 nanomaterials-14-00657-f003:**
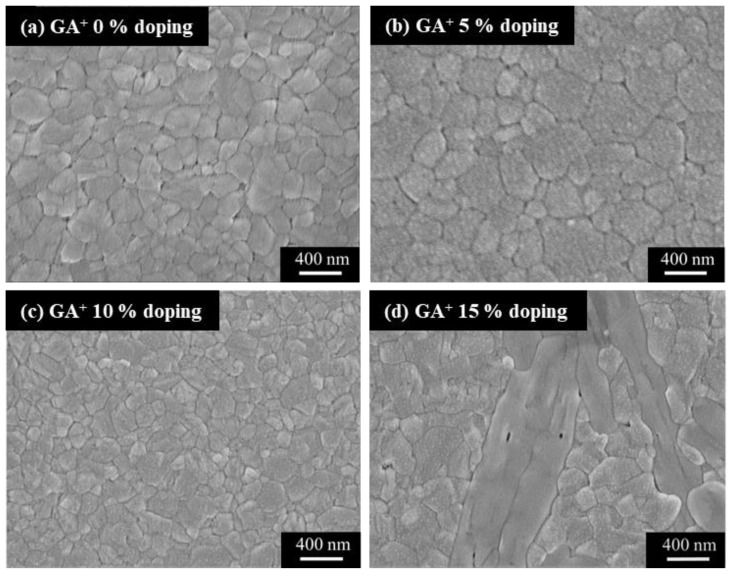
SEM images of (GA)_x_(MA)_1−x_PbI_3_ films with various doping concentrations of (**a**) 0, (**b**) 5, (**c**) 10, and (**d**) 15%.

**Figure 4 nanomaterials-14-00657-f004:**
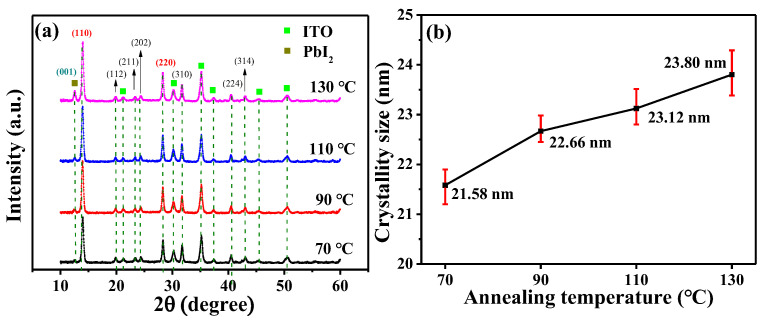
(**a**) XRD patterns and (**b**) relating crystal grain size of (GA)_0.05_(MA)_0.95_PbI_3_ films treated with various annealing temperatures.

**Figure 5 nanomaterials-14-00657-f005:**
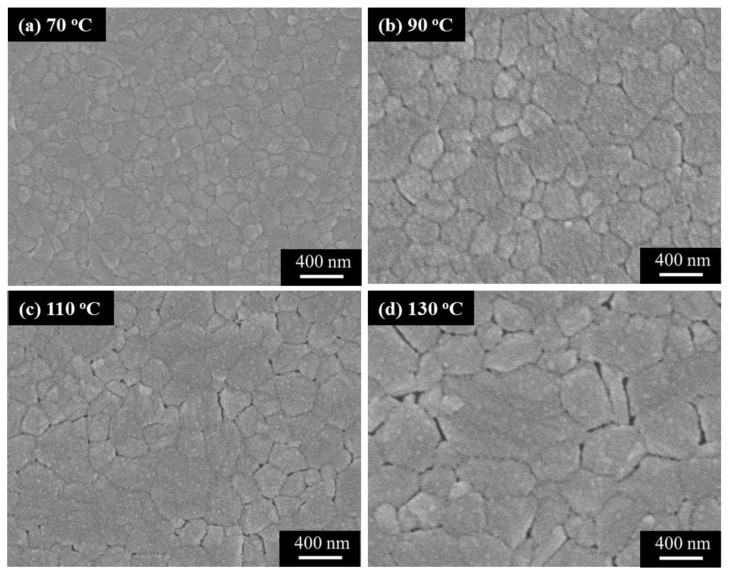
SEM images of (GA)_0.05_(MA)_0.95_PbI_3_ films treated with various annealing temperatures of (**a**) 70, (**b**) 90, (**c**) 110, and (**d**) 130 °C.

**Figure 6 nanomaterials-14-00657-f006:**
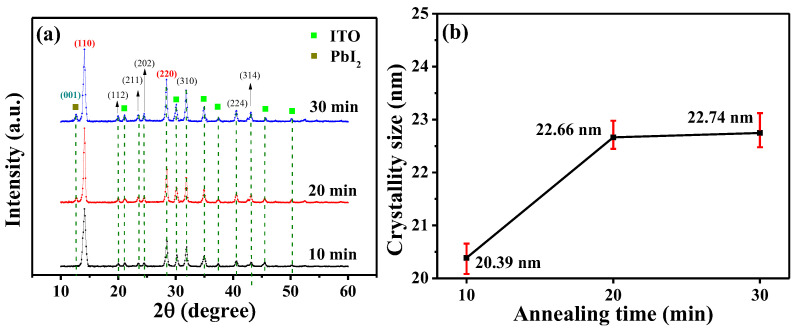
(**a**) XRD patterns and (**b**) relating crystal grain size of (GA)_0.05_(MA)_0.95_PbI_3_ films annealed for various times.

**Figure 7 nanomaterials-14-00657-f007:**
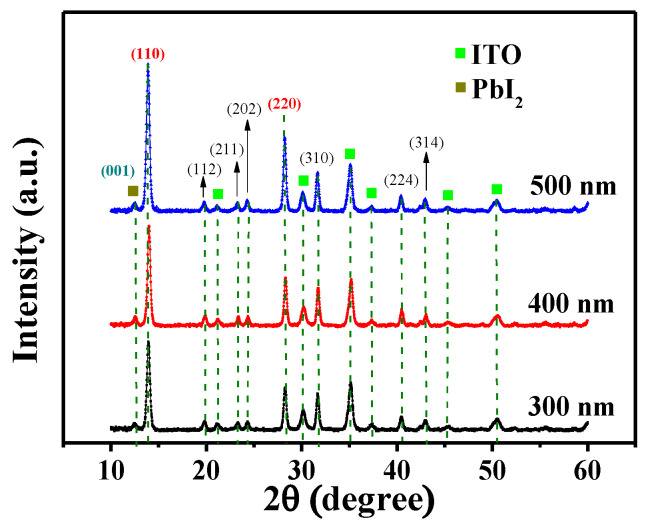
XRD patterns of (GA)_0.05_(MA)_0.95_PbI_3_ films with various thicknesses.

**Figure 8 nanomaterials-14-00657-f008:**
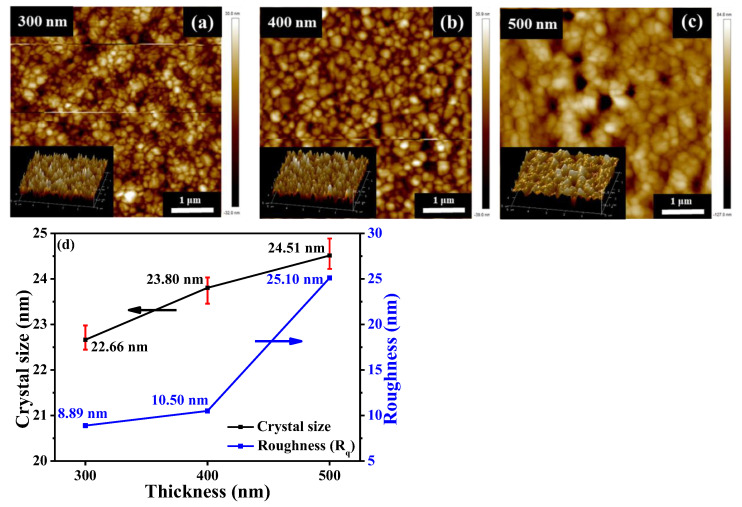
AFM images of (GA)_0.05_(MA)_0.95_PbI_3_ films with various thicknesses of (**a**) 300, (**b**) 400, and (**c**) 500 nm. (**d**) Relating crystal grain size and roughness of (GA)_0.05_(MA)_0.95_PbI_3_ films with various thicknesses.

**Figure 9 nanomaterials-14-00657-f009:**
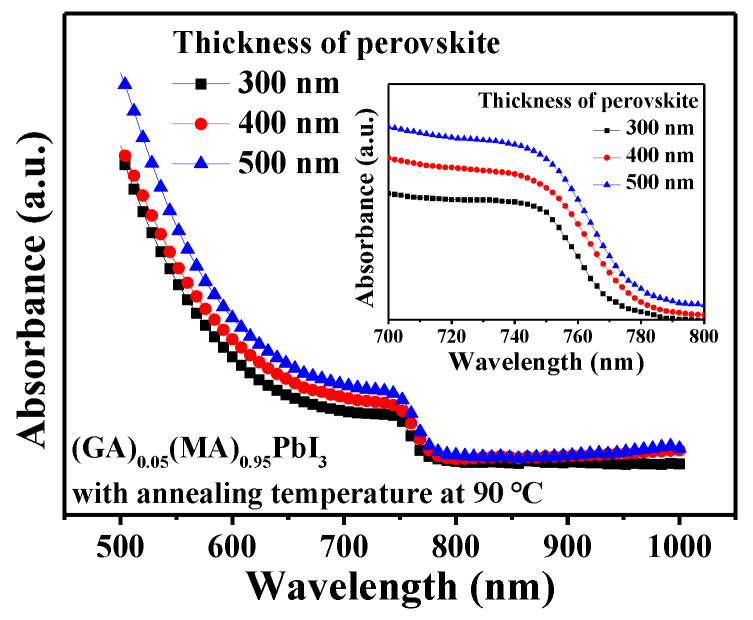
Absorption spectra of (GA)_0.05_(MA)_0.95_PbI_3_ films with various thicknesses. Inserted figure shows extended absorption spectra between wavelength of 700 and 800 nm.

**Figure 10 nanomaterials-14-00657-f010:**
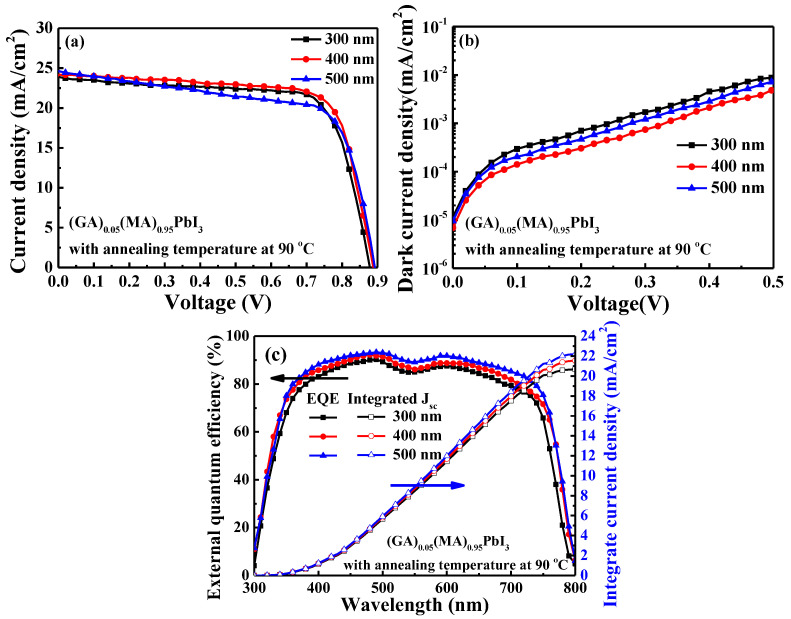
(**a**) Current density-voltage, (**b**) dark current density-voltage, (**c**) external quantum efficiency and integrated current density characteristics of PSCs using (GA)_0.05_(MA)_0.95_PbI_3_ perovskite active layer with thicknesses of 300, 400, and 500 nm.

**Figure 11 nanomaterials-14-00657-f011:**
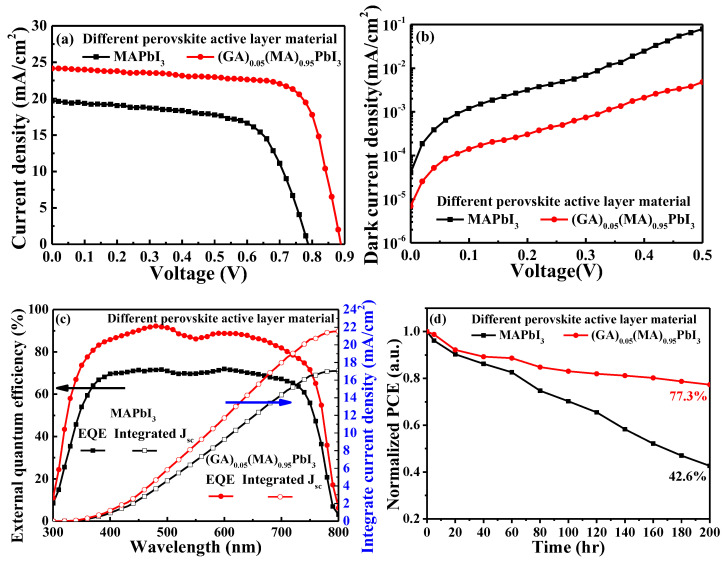
(**a**) Current density-voltage, (**b**) dark current density-voltage, (**c**) external quantum efficiency and integrated current density characteristics, and (**d**) 200-h stability test of PSCs with MAPbI_3_ perovskite active layer and with optimal (GA)_0.05_(MA)_0.95_PbI_3_ perovskite active layer.

**Table 1 nanomaterials-14-00657-t001:** Characteristics of PSCs with various thicknesses of (GA)_0.05_(MA)_0.95_PbI_3_ perovskite active layers.

Perovskite Thickness (nm)	V_oc_ (V)	J_sc_ (mA/cm^2^)	FF (%)	PCE (%)	Integrated J_sc_ (mA/cm^2^)
300	0.879	23.88	73.1	15.36	20.69
400	0.891	24.21	73.1	15.78	21.55
500	0.894	24.75	66.6	14.73	22.21

**Table 2 nanomaterials-14-00657-t002:** Characteristics of PSCs with MAPbI_3_ perovskite active layer and with optimal (GA)_0.05_(MA)_0.95_PbI_3_ perovskite active layer.

Perovskite Active Layer	V_oc_ (V)	J_sc_ (mA/cm^2^)	FF (%)	PCE (%)	Integrated J_sc_ (mA/cm^2^)
MAPbI_3_	0.783	19.71	65.0	10.03	17.05
(GA)_0.05_(MA)_0.95_PbI_3_	0.891	24.21	73.1	15.78	21.55

## Data Availability

The data presented in this study are available upon request from the corresponding author.

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
