# Peer review of "Investigation of Perovskite Solar Cells Using Guanidinium Doped MAPbI3 Active Layer"

_nanomaterials, 2024, doi:10.3390/nano14080657_

Round 1

Reviewer 1 Report

Comments and Suggestions for Authors

In this manuscript the authors show that adding 5% of GA+ into the MAPbI3 perovskite, not only the performance but also the stability of the device is improved. They checked different GA+ concentrations and they claimed that the 5% is the optimal one looking at the crystallinity and surface morphology. Also, they tested different annealing temperatures, being 90ºC the best for the perovskite. Moreover, they prepared devices with three different thicknesses from the optimal perovskite, with an enhanced stability for the 400 nm perovskite device.

Evaluation

I would publish this paper after major revision.

From my point of view, XRD and surface morphology is not enough analysis for discarding a perovskite. The one of 10% of doping has similar characteristics than the 5% doping. For this reason, I would prepare devices with the 10% doping perovskite with the best perovskite thickness and annealing conditions.

 Moreover, I would add devices without GA+ to show, indeed, that the GA is doing something on the efficiency.

Statistics would be need it, to show that the devices with GA are reproducible.

Questions:

1. You checked different annealing temperatures, but did you check also different annealing times? And why did you choose to anneal 20 min?

2. Why the current extracted from the EQE is much lower than the current extracted from JV curves? 22.21 mA cm-2 vs 24.75 mA cm-2

3. According to the perovskite bandgap, Voc should be much higher than the one obtained in the three perovskites. Why is the Voc so low in the three devices?

4. Are the devices used for measuring stability encapsulated?

5. You did not show up any statistics, is this improved stability reproducible?

6. You claimed: “According to the experimental results, by doping GA+ into MAPbI3 perovskite films, not only improved performances but also boasted long-term stability of the resulting (GA)0.05(MA)0.95PbI3 films”. However, I went through the whole manuscript and there are no results of the devices with the perovskite without GA+. How do we know that with the GA+ (5 %) there is an improvement of the performance? Some results are missing to be able to claim that the devices with GA are better in terms of efficiency.

Comments on the Quality of English Language

Spelling mistakes

Line 80- “spun”

Line 88- “wer e”

Reviewer 2 Report

Comments and Suggestions for Authors

Perovskite solar cells are among the most popular fields of study in modern chemistry. Among other formulations of active layer organic semiconductors with an ABX3 lattice structure are one of the most promising as they allow a wide variation of composition leading to construction of devices with high carrier mobility, tunable band gap, high power conversion efficiency. However, stability of organic perovskites under exposure to oxygen and humidity remains crucial factor. This problem can be solved by doping of active layer by different additives, i.e., guanidinium. In the paper under review the authors have studied effect of guanidinium content and synthesis conditions on the properties of perovskite solar cells to find the optimal conditions of their fabrication.

The authors carried out very comprehensive experimental studied. Different factors have been analyzed and allow them to recommend a 400-nm-thick (GA)x(MA)1-xPbI3 film, with 5% GA+ doping annealed at 90 °C as the optimal kind of active layer for perovskite solar cells fabrication. I would like to emphasize that authors not only mentioned the difference in properties of layers obtained under different conditions but gave some explanations of the observed tendencies (unfortunately, this very valuable approach is not common one).

I recommend this manuscript for the publication but I would like to pay attention of authors to some minor drawbacks:

Line 33. Perovskite can contain inorganic cation as well. I guess, it is better to write: “Organic perovskites, a type of organic semiconductors…”

Lines 45-60. I recommend to support the literature review by specific figures.

Figures 4b and 7b. Grain sizes calculated according to Scherrer equation are presented. (1) What is the accuracy of the calculations? Is that really 23.12 nm or maybe 23.15 nm? (2) How many parallel synthesizes have been made and analyzed or that is the data for the unique synthesis? I also recommend to use uncertainty bars at Figures.

Round 2

Reviewer 1 Report

Comments and Suggestions for Authors

It can be published as it is, no comments for the authors